# The Difference in the Creativity of People Who Are Deaf or Hard of Hearing and Those with Typical Hearing: A Scoping Review

**DOI:** 10.3390/children10081383

**Published:** 2023-08-14

**Authors:** Petra Potměšilová, Miloň Potměšil, Miloslav Klugar

**Affiliations:** 1Department of Christian Education, Sts Cyril and Methodius Faculty of Theology, Palacký University, 77900 Olomouc, Czech Republic; petra.potmesilova@upol.cz; 2Center of Evidence-Based Education & Arts Therapies: A JBI Affiliated Group, The Institute of Special Education Studies, Faculty of Education, Palacky University Olomouc, 77147 Olomouc, Czech Republic; 3Czech National Centre for Evidence-Based Healthcare and Knowledge Translation (Cochrane Czech Republic, Czech EBHC: JBI Centre of Excellence, Masaryk University GRADE Centre), Faculty of Medicine, Masaryk University, 62500 Brno, Czech Republic; klugar@med.muni.cz

**Keywords:** hard of hearing persons, person, deaf, creativeness, review, academic, thinking

## Abstract

The scoping review aimed to describe differences in creativity between deaf and hard of hearing and typically hearing people. The research question for the review was: what are the differences in the creativity of deaf and hard-of-hearing people in comparison with people with typical hearing? A total of eleven databases were used for the search, as well as sources of the unpublished studies/gray literature. The scoping review was prepared following the Joanna Briggs Institute methodology and PRISMA frame as a basis for reporting scoping reviews. A total of 30 studies were analyzed concerning the selected research areas. Intrinsic creativity was the first area identified. Specific activities for the development of creativity formed the second area for analysis. The third area focused on differences in creativity between deaf and hard of hearing and typically hearing. The fourth area includes studies that call for an equitable research environment.

## 1. Introduction

Human creativity is viewed as a talent that is considered a source of success with an impact on human society [1]. In the history of human society, appreciation has been given to creative problem solving or the development of completely new things in all areas of society, including culture [2,3]. The individual aspect of creativity is marked by the ability to view the current reality from an unconventional perspective and thus bring new and unconventional solutions. Creativity is also considered to be an important manifestation of a person’s intellectual functioning. For the learning process, both in schooling and as an important part of human life, creativity is one of the important supporting factors. The level of creativity of an individual can be assessed using performance measures derived from creative thinking [4]. Creative thinking skills include finding, analyzing, and solving problems from different perspectives that have not been identified by others [5]. 

The theory of the connection between intellect and creativity was developed by Guilford [6] in his concept of the structure of intellect. Within the structure of the intellect, he distinguishes five operations—cognition, memory, convergent thinking, divergent thinking, and evaluation. Creativity can be conceptualized as the drive and skill to generate new ideas or alternatives to solve problems, develop forms of communication, and ultimately develop personal life in a broader perspective. Thinking, which among other things leads to problem-finding and the search for solutions, is assessed as the basic and highest monetary function, and this is precise with the support of creativity. In the broadest view of the application of creativity, it can be said to be the driving force behind a process that produces new knowledge and original approaches. All is necessary not only for the development of the individual but also for the development of human progress. Amabile and Pillemer [7] consider creativity as a form of skill to teach, practice, and shape. With this statement, we can turn to the process of education and the school work of children. Guilford’s contribution lies mainly in the distinction between two types of productive thinking—convergent and divergent. Convergent thinking refers to the situation of the conventional or only possible solution. The person knows where to go but does not know the course of action. In contrast, divergent thinking refers to the search for new ways of solving especially in situations with multiple possible solutions. This is presented in the context of mathematics teaching by Bingölbali and Bingölbali [8]. A person seeks both a goal and a course of action. They are interdependent, although the degree of dependence decreases as the intelligence quotient (IQ) increases [9]. 

Deafness and hard of hearing (DHH) are manifested by differences in communication and the construction of communication competencies. DHH is considered to be one of the most serious handicaps that can influence an individual’s development, depending on the type and level [10,11,12,13]. Especially from the educational point of view, this disability can have a major effect on the person concerned, in particular as regards communication and thinking capacities. According to the WHO [14], it is possible to define three basic areas of impact of DHH: the functional, social and emotional, and economic areas. DHH has a fundamentally negative impact on the development of vocabulary and understanding of verbal terms, especially if they are not wholly concrete, which means the saturation of the concept bank, activation of the tongue, and also its use in communication [15,16,17]. Marschark et al. [18] suggest that the academic achievement of DHH students is the result of the complex interplay of many factors. These factors include characteristics of the students (e.g., hearing thresholds, language fluencies, mode of communication, and how their communication functions), characteristics of their family environments (e.g., parents’ level of education, socioeconomic status), and experiences inside and outside school (e.g., school placement, having been retained at a grade level). DHH children who grow up in a language-less environment have fundamental problems in the construction of mathematical ideas and thus success in school mathematics and generally speaking in school success [19,20]. 

For the education and training of people with hearing impairments, the basis for process education is the same as for hearing people. In the current approach to the education of children and pupils with hearing impairments, it is the merit of effective communication through sign language that enables creativity to develop [21,22]. For this research, the most important impact is the functional one, which means the effect of DHH on the creation of communication skills that can influence educational possibilities.

This review aimed to describe the differences in creativity between deaf and hard-of-hearing people and people with typically hearing.

## 2. Materials and Methods

The scoping review focusing on the creativity of DHH people provides significant information that can be used in research and practice in the fields of education, psychology, or culture. In this study, MEDLINE, the Cochrane Database of Systematic Reviews, Epistemonikos, and The Joanna Briggs Institute (JBI) Evidence Synthesis were used for the search. Following the JBI guidelines, the scoping review brings together all types of evidence available for the given topic. The results of the review provide information about gaps in the research and thus form a menu of research opportunities for the scientific community [23,24].

### 2.1. Review Question

What are the differences in the creativity of deaf and hard-of-hearing people in comparison with people with typical hearing?

### 2.2. Eligibility Criteria

Eligibility criteria were defined based on the JBI methodology [24]: types of participants, concept, and context [25]. Types of sources of evidence were defined after them. 

#### 2.2.1. Participants

This scoping review will look at the creativity of people with DHH. For this study, the term ‘deaf and hard of hearing’ refers to carriers of a disability that is congenital or acquired before the age of two, as well as prelingual deafness and those who have received special education intervention. Therefore, individuals who are DHH after speech development, individuals with cochlear implants (CI), and those who did not need any special education intervention will not be included in the review. There is no age restriction for included participants.

#### 2.2.2. Concept

The scoping review will consider studies that explore creativity in DHH people. The scope is to describe the specifics of creativity in deaf and hard-of-hearing people and to identify possible differences from people with TH. 

#### 2.2.3. Context

This review will consider studies from schools and counseling and therapeutic settings, which include both people with TH and DHH people.

### 2.3. Types of Sources of Evidence 

This scoping review considered both experimental and quasi-experimental study designs, including randomized controlled trials, non-randomized controlled trials, before and after studies, and interrupted time-series studies. In addition, analytical observational studies, including prospective and retrospective cohort studies, case-control studies, and analytical cross-sectional studies were included. This review was accepted for inclusion in descriptive observational study designs, including case series, individual case reports, and descriptive cross-sectional studies.

Qualitative studies that focus on qualitative data, including, but not limited to, designs such as phenomenology, grounded theory, ethnography, qualitative description, action research, and feminist research were included. Mixed-methods research designs were considered as well. 

In addition, systematic reviews that meet the inclusion criteria were included in this scoping review. 

Text and opinion papers were also included in this scoping review. There were no restrictions regarding the year of publication and language of publication (if the studies had at least a title and abstract in English); studies published from the inception of the database to the present were included, as they may be relevant regardless of the publication date.

### 2.4. Methods 

The scoping review was prepared following the JBI methodology [24], which states the following steps: Search strategySource of evidence screening and selectionData extractionAnalysis and presentation of results

The protocol of the scoping review has been published as a priori [25]. There are no deviations from the protocol in the full scoping review. 

#### 2.4.1. Search Strategy 

The search strategy aimed to locate both published and unpublished studies. An initial limited search of Ovid MEDLINE(R) 1946 and CINAHL Plus with Full Text was undertaken to identify articles on the topic. 

The words contained in the titles and abstracts of relevant articles and the index terms used to describe the articles were used to develop a full search strategy. The search strategy, including all identified keywords and index terms, was adapted for each database and/or information source that is included. The databases that were searched included ProQuest Central, Web of Science Core Collection, Scopus, PsycArticles, PsycINFO, MEDLINE (OvidSP), Annual Reviews, EBM Reviews, and CINAHL Plus with Full Text. Sources of the unpublished studies/gray literature, including Clinical trials, Current controlled trials, CENTRAL, and Google Scholar, were also searched (see Appendix A).

#### 2.4.2. Screening and Selection of Sources of Evidence 

Following the search, all identified citations were collated and uploaded into EndNote 20/2021(Clarivate Analytics, Philadelphia, PA, USA) and duplicates were removed. Following a pilot test, titles and abstracts were screened by two independent reviewers (P.P., M.P.) for assessment against the eligibility criteria for the review. Potentially relevant sources were retrieved in full, and their citation details were imported into the JBI System for the Unified Management, Assessment and Review of Information (JBI SUMARI) [26]. The full texts of selected citations were assessed in detail against the eligibility criteria by two independent reviewers (P.P., M.P.). Fourteen papers were excluded after the full-text screening. The reason for exclusion was a different concept that did not meet the inclusion criteria. The third reviewer (M.K.) took part in the discussion about the Relevancy of Text and Opinion papers. 

The results of the search and the study inclusion process are reported in full and presented in a Preferred Reporting Items for Systematic Reviews and Meta-analyses extension for scoping review (PRISMA-ScR) flow diagram [27] (see Figure 1).

#### 2.4.3. Data Extraction 

The data extraction tool was created by the authors and contains three basic areas, which are then divided further:General study details (authors, publication date, country, respondents);Methodology (characteristics of creativity, study design, tools, area of interest, and study aim);Output (1. description of creativity; 2. possibilities of development of creativity; 3. differences between DHH and TH, internal validity and reliability of tools developed) and verifiability of research and tools.

The three categories (see above) correspond to the research question of the scoping review. Before answering this question, it was necessary to describe the area of creativity of an individual who is DHH. 

#### 2.4.4. Analysis and Presentation of Results

Based on the consensus of all three reviewers, tables were prepared in which individual areas are presented on the basis of the data extraction tool. The basic analysis is presented in the text; detailed information is presented in the tables to which reference is made.

## 3. Results

### 3.1. General Study Details

The extracted articles were written between 1942 and 2019. The focus of research interest on the creativity of individuals with DHH was perhaps at its peak in the 1970s and 1980s (16 out of 30 studies). Twenty-three studies were conducted in the USA, two in Central Europe (Poland and Slovakia), the remainder in the People’s Republic of China, Nigeria, and India, and two in Indonesia. In 10 cases, the research population consisted of DHH individuals only, in 19 cases the research population was a comparison of DHH respondents with the TH population (see Table 1 for details), and in one case [28] the creativity of TH and visually impaired individuals was examined. Overall, the number of respondents ranged from 1 to 777. For three studies, the number of respondents was not given. In 9 cases, the gender of the respondents was not stated, and in the remaining 21 cases, the number of women (girls) and men (boys) was explicitly stated. In all these cases, the authors of the studies tried to maintain a balanced gender ratio of respondents. In three cases, the age of the respondents was not given. In the other studies, the ages of the respondents ranged from four to 88 years; the focus of the research was on the population from school age to adolescence. The hearing loss of the respondents was clearly defined only in some cases; where no specific value loss appeared, a note type was given: profoundly deaf, i.e., they did not benefit from hearing aids.

### 3.2. Methodology 

#### 3.2.1. Characteristics of Creativity

The first important item that was analyzed for all the selected articles was the characterization of the concept of creativity. In nine articles [29,31,35,45,46,47,48,49,51], it is not explicitly stated which theory of creativity the authors are drawing on. Daramola et al. [30] provide a characterization ability to perceive the world in new ways to find hidden patterns, make connections between seemingly unrelated phenomena, and generate solutions. In other cases, the authors draw on Torrance’s theory of creativity. Torrance [56] characterizes creativity as a complex of abilities, among which he includes sensitivity to problems—the ability to perceive and search for problems, fluency—the ability to produce a multitude of ideas, flexibility—the ability to have a variety of perspectives, originality—the ability to have a new and different perspective, elaboration—the ability to be careful and sophisticated, and redefinition—the ability to define a problem unusually. Ebrahim [31] Laughton [42] and Stanzione et al. [22] then add to Torrance’s notion of creativity with the definition of thinking of Guilford [57], who distinguished two types of thinking, namely convergent and divergent. The relationship between fluency and verbal creativity is based on the close connection between intelligence and creativity. Creativity is characterized according to the Cattell–Horn–Carroll concept of intelligence as a significant factor at a similar functional level to the fluency of thought and verbal fluency [58,59]. 

Divergent thinking allows for a search for different alternatives in the context of problem-solving, from which the most appropriate one is then selected using convergent thinking. According to Guilford [57], divergent thinking, which is closely related to creativity, has the following components: fluency or the smooth flow of ideas, flexibility or flexibility of thought, originality, sensitivity to problems, redefinition or the ability to use prior knowledge in a new way, and elaboration, i.e., the creation of functional details in problem-solving. 

#### 3.2.2. Study Design 

Depending on the chosen design, the articles can be divided into three basic groups (see Figure 1): quantitative, mixed, and Text and Opinion Papers.

##### Quantitative Design

Most of the articles (19 of 30) describe quantitative research. As can be seen from Table 2, the quasi-experimental research design was used in 15 articles, case series in three, and before and after studies in one. In the case of quasi-experimental research design, except for two articles [41,52], DHH and TH subjects were compared. Kaltsounis [41] looked at the effect of race concerning DHH, and Reber and Sherrill [52] compared the effect of dance training in two DHH groups.

In two cases [49,51] the area of interest for exploring creativity was not explicitly stated, while the remaining 17 were about the potential for using the findings in teaching. Of the 19 articles, 14 dealt directly with the relationship between the creativity of children with DHH and children with TH. Of these 14 studies, 4 [37,43,49,51] related the comparison of creativity to other factors such as intelligence, motor skills, or social relationships in the classroom. The remaining five cases involved different characteristics of creativity in individuals with DHH. The authors of two articles [42,52] considered the possibilities for the development of creativity in DHH individuals, and in three cases [34,36,41] creativity in DHH individuals was examined as a separate phenomenon with a focus on intelligence or language ability. The Torrance Tests of Creative Thinking were used to determine levels of creativity in 11 cases, supplemented in one case [50] by the Barron–Welsh Art Scale and in another [52] by the Dance/Movement Skills Assessment. In the remaining cases, there were different tests of creativity chosen by the researchers concerning the issue at hand (see Table 2 for details), and in one of these cases [30] the researchers developed their own instrument. Dramola et al. [30] reported that the instrument they created was validated by experts in the Unit of Educational Research, Measurement, and Evaluation, Department of Social Sciences Education, Faculty of Education, Loyola University, and the reliability of the instrument would be determined using a split-half method (Cronbach’s α = 0.76).

##### Mixed Design

A mixed design was used in three articles. In all three cases, it was a comparison of creativity in DHH and TH children. In two cases [46,47], categories through which the specifics of creativity in DHH individuals were described were developed sequentially. In one case [53], levels of originality, fluency, flexibility, and elaboration were described for individuals with DHH. Qualitative analyses were supplemented in all cases by descriptive analysis. The relationship between creativity in children with DHH and children with TH was then compared using correlations (see Table 3).

In all three cases, the research results were directed toward the school environment. In two of the cases [46,47], the effect of language and cognitive level on creativity was investigated in comparison to TH peers. Silver [53] examined the relationship between cognitive functions and creativity, again in comparison with TH peers. Silver [53] used the Torrance Tests of Creative Thinking. Marschark and West [46] and Marschark et al. [47] chose their instruments: the respondents were asked to tell two stories each time. In both studies, one story was the same and the other was different. The common story is that once while walking through the mountains, they came upon a hidden door and a staircase leading deep into the earth. Below, they found a large cave and a previously unknown civilization. Different stories: They awoke one day to discover that animals and people had exchanged roles. They were asked to tell about seeing someone climbing a ladder and entering the window of a neighbor’s house (see Table 3).

##### Text and Opinion Studies

A total of eight articles that could not be classified in the previous categories because of missing or insufficient methodology were included in this group. In addition to being articles that met the inclusion criteria, these are articles that provide substantial information on the issue at hand. 

Three studies [29,31,48] were concerned with describing different activities that can lead to the development of creativity in individuals with DHH. These include various activities in mathematics, drama, or art. In all these cases, the authors of the articles point out that the influence of specific activities can lead to the development of creativity. One article [38] briefly describes various teaching methods for DHH individuals and compares them in terms of their impact on creativity. Another study [28] is a brief report comparing the creativity of individuals with DHH and those with visual impairments. Hicks [35] compares creativity in DHH and TH individuals. However, he adds that regarding sample size, the results of the study cannot be generalized. The last two articles are different. Marschark and Clark [44] produced a literature review whose purpose was to review and weigh the available evidence concerning the non-linguistic and linguistic creative abilities of DHH children. Marschark et al. [45] presented information on a newly developed instrument for examining creativity in individuals with DHH. At the same time, they stress the need to approach individuals with DHH individually and to create a research environment for them that does not discriminate against them.

#### 3.2.3. Outputs

The last area to be analyzed was research outputs. The research outputs were divided into three categories: Description of creativity,Development of creativity,Differences in creativity between HDD and TH people.

Each area also includes information on the verifiability of the research and the instruments used (see Table 4).

Potential differences emerge, but mainly it is essential to create an appropriate testing environment. Under current conditions, underestimation occurs (Table 5 and Table 6).

## 4. Discussion

The scoping review aimed to gather information on all available research in the field of the creativity of individuals with DHH. Specifically, we were interested in the characteristics of creativity in individuals with DHH, and whether and how the creativity of individuals with DHH differs from that of individuals with TH.

On the basis of an analysis of 30 studies, it was possible to identify four distinct areas in the cognition of creativity in individuals with DHH. The first area to be identified is self-reported creativity. The sources reviewed suggest that the creativity of individuals with DHH may be different. In this case, it is not a matter of assessing whether or not it is ‘more creative’, but that it is different. Hicks [35] states the following characteristic differences in the work of DHH individuals: ‘where children with TH draw animals or toys, DHH individuals choose buildings or nature’. Hicks [35] then sees further differences in the different ways creations are described—a greater need to copy and explain.

The second area identified is specific activities for developing creativity. A total of seven studies (see Table 5) focused on this area. Arnidha and Hidayatulloh [29] and Kaltsounis [38] conclude that it cannot be proven that specific activities can lead to the development of creativity in individuals with DHH. However, others [31,36,42,48,52] reach the opposite conclusion. In their research, they point out that appropriately chosen activities have been shown to have a good effect on the development of creativity in individuals with DHH.

The third area is the differences in creativity between individuals with DHH and individuals with TH. This area consists of studies that have looked at areas in which individuals with DHH are more or less creative than individuals with TH. Eight articles present research findings that suggest individuals with DHH are more creative than their peers with TH [23,28,30,37,39,40,46,53]. All agree that individuals with DHH have higher fluency scores. Ten studies [32,33,36,43,49,50,51,54,55] suggest the opposite, namely that individuals with TH are more creative compared to individuals with DHH. The authors agree this is refer to a higher level of creativity overall, it is referring to statistically significant for originality, flexibility, and especially in the area of verbal originality. The authors agree that individuals with TH can generalize and name creations in the abstract, whereas individuals with DHH resort more to description and narration. In the microstructure of the brain, differences in some pathways for language-related functions localized mainly in the left arcuate fasciculus have been demonstrated between deaf individuals growing up in a language-free environment and those exposed to language from early childhood. It was shown that native sign language speakers (i.e., especially DHH children of DHH parents) had the same connectivity in these brain pathways as hearing people who communicated in spoken language. This confirms the massive negative influence of the language-free period and in turn, highlights the positive influence of early language and communication experiences in infancy [60].

The fourth and final area is made up of studies that call for the creation of an equal research environment. Marschark and Clark [44] and Marschark et al. [45] stress the need to create an appropriate environment for researching individuals with DHH. If these conditions are not created, individuals with DHH are undervalued. Marschark et al. [47] are the only ones to state that if appropriate conditions are created, the creativity of children with DHH is comparable with that of those with TH.

Creativity problems in deaf people may be related to their limited access to auditory information. Hearing loss can affect their perception and understanding of music, sounds, and speech, which can affect their ability to express themselves through these media. Deaf people may also face communication barriers that may prevent them from sharing their thoughts, ideas, and perceptions [61]. However, it is important to note that creativity is not limited to the auditory dimension. Many forms of creativity do not require auditory perception. Creativity is a universal ability and can manifest itself in many different ways, regardless of whether someone has a hearing impairment [22]. Deaf people may exhibit the same mechanisms of creativity as people with typical hearing. It is possible to consider the basic mechanisms of creativity that may be specific to deaf people: 1. Visual sense—Deaf people often have a more strongly developed visual sense to compensate for their lack of hearing. They can therefore be very creative in visual art, design, photography, or creating visually appealing concepts [62]. Language and Communication—Deaf persons can be creative in language and communication, both written and visual. They may create original and innovative ways of expressing their ideas using sign language, pictograms, or other visual media. Some deaf individuals have also adopted sign language, a language of communication, allowing them to share their thoughts and ideas through visual communication [46]. Technology and innovation—Deaf people often use technology and innovation to improve their quality of life. They may be involved in the development of new technologies, apps, or devices that aid communication or auditory rehabilitation [63]. Experience and perspective—Deaf people may have unique experiences and perspectives that they can bring to the creation of art, literature, or film [64]. Their skills and perspective can make them a valuable asset in the creation of collective projects such as theatre, film, or design [65]. It is important to remember that each person is unique and may exhibit different mechanisms of creativity regardless of their hearing impairment. Deaf people have the potential to be very creative and their unique perspective on the world can bring new and innovative ideas to different areas of art, technology, and communication. It is important to support and respect the individual needs and experiences of deaf people, and to encourage their creativity in different areas.

### 4.1. Limitations of the Studies Included

There are two main limitations of the studies included. The first is the date of publication of the articles. Only 6 articles out of 30 were published after 2000. The second limitation is that while some of the research provides important information, it is not possible to obtain reliable information about how the research was conducted or to determine the validity and reliability of the instrument used or the validity of the research.

### 4.2. Limitations and Strengths of This Scoping Review

The strengths of this scoping review can be considered to be the uniqueness of the issue under investigation. No scoping report of this type has been identified based on the above literature search. The aforementioned study by Marschark and Clark [44] can be considered a first attempt, but it remains at the level of a literature review without in-depth analysis. Another strength of the present scoping review is the identification of potential research directions. 

The weakness of the scoping review can be considered to be the sporadic nature of the issue under investigation. It may seem to be a marginal and unimportant area. The learning environment has a great effect on the cultivation of creative thinking [66]. However, according to Aliotti and Blanton [67] or Gajda [68], it is clear that creativity can have an impact on school achievement, and therefore this issue can be considered essential in the search for optimal and equitable conditions for DHH education.

## 5. Conclusions

Based on a systematic search strategy, a total of 30 articles were identified that dealt with the relationship between creativity and individuals with DHH. The articles analyzed revealed that there are differences between the creativity of individuals with DHH and the creativity of individuals with TH. The researchers see a major difference in verbal creativity, in which individuals with TH score better, whereas individuals with DHH score higher in fluency. From the above, it can be assumed that the difference in verbal creativity is due to differences in language acquisition and competence in the two groups compared. The researchers also point out that it is necessary to create appropriate conditions for examining the creativity of individuals with DHH (e.g., the use of sign language) so that creativity can be assessed objectively and individuals with DHH are not underestimated. The Torrance Tests of Creative Thinking appear to be an appropriate tool for examining creative thinking. This test allows for a comprehensive examination of creative thinking and appears to be objective even in the case of individuals with DHH.

Creativity is an important manifestation of human existence with the need for self-expression. Brain activity and the pathophysiology of hearing are linked in a way that they influence each other with an impact on the function of creativity. Various studies have shown that individuals with certain brain disorders or hearing impairments may experience changes in their abilities, including creative abilities [69]. Brain pathophysiology may excel in fields such as music or art, but their overall creative thinking may be limited due to difficulties in social interaction and communication [70]. Similarly, the pathophysiology of hearing can affect an individual’s creativity. People with hearing impairments may have difficulty accurately perceiving auditory stimuli, which may limit their ability to appreciate music or understand the nuances of spoken language. This limitation can restrict their exposure to different forms of artistic expression and limit their creative potential. However, it is important not to generalize these findings to all individuals with brain or hearing pathophysiology [71]. Many individuals with these conditions have demonstrated remarkable creative abilities despite their challenges [22]. In addition, advances in technology and assistive devices have enabled individuals with hearing impairments to access auditory information more effectively. In conclusion, although the pathophysiology of the brain and hearing can potentially affect human creativity by limiting certain aspects of artistic expression, it is crucial not to underestimate the resilience and adaptability of individuals who face these challenges. By providing appropriate support systems and adaptations, society can help unlock the full creative potential of each individual, regardless of any underlying condition [72].

## 6. Implication for Future Research

Given the above findings and the limitations mentioned, it would be useful to conduct primary research in two areas. The first area would be to measure the detailed characteristics of the creativity of individuals with DHH. The target of this research could be alternative norms for the Torrance Tests of Creative Thinking for individuals with DHH. In the case of qualitative research, this would be a grounded theory study and in the case of quantitative research, it would be an experimental research design. The second area could be research that looks at the differences in creativity between individuals with DHH and those with TH. In this case, it could be quantitative research (randomized controlled trial). the basic research questions could then be: What are the characteristics of creativity in individuals with DHH? What tools can be used to objectively determine the level of creativity in individuals with DHH? How do the creativity levels of individuals with DHH and those with TH differ?

However, it would be essential for the research to be conducted under non-discriminatory conditions: the researcher should be an expert in the field of DHH and the instructions given should be in line with the principles of communication with individuals with DHH (sign language, adapted text).

## Figures and Tables

**Figure 1 children-10-01383-f001:**
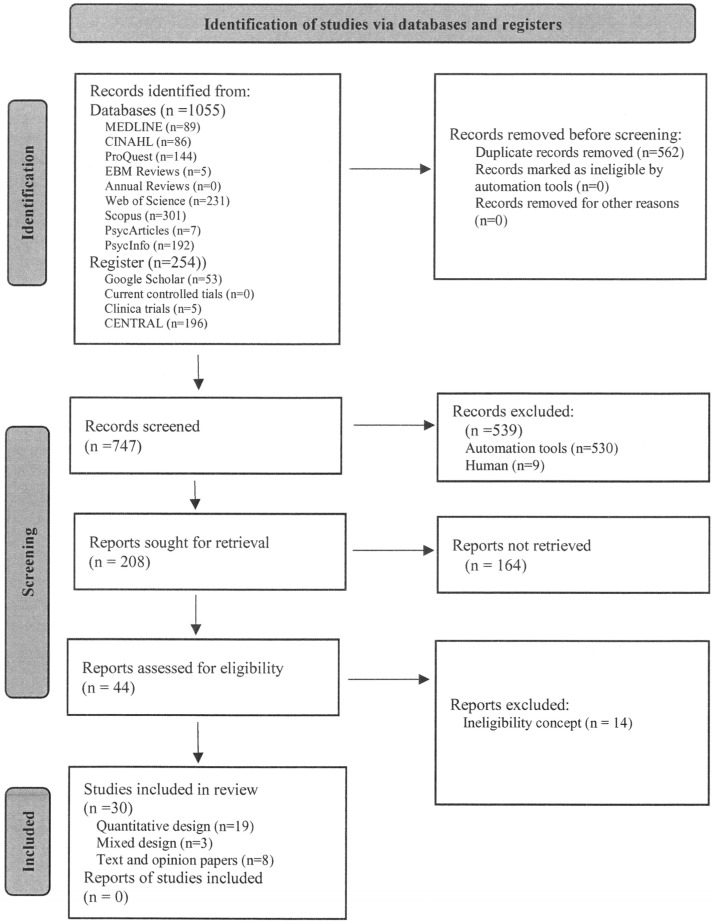
Prisma flow diagram.

**Table 1 children-10-01383-t001:** General study details.

Authors and Publication Date	Country	Respondents			
		Number N (DHH)	Gender	Age	Characteristic of DHH
Arnidha and Hidayatulloh, 2019 [29]	Indonesia	5	NS	fifth grade	deaf
Daramola et al., 2019 [30]	Nigeria	248 (146)	NS	second year	NS
Davies, 1984 [31]	United States of Amerika	NS	NS	NS	Hearing-impaired
Ebrahim, 2006 [32]	United States of Amerika	410 (210)	NS	8–11 y	Hear loss 90–131 dB
Ebrahim, 2006 [33]	United States of Amerika	410 (210)	NS	8–11 y	Hear loss 90–131 dB
Gallagher, 1968 [34] *	United States of Amerika	74	Boys 34 Girls 40	4th–8th grades	deaf
Halpin and Torrance, 1973 [28]	United States of Amerika	68 (34)	NS	9–11 y	Hearing deprivation was substantial enough to prevent them from making satisfactory progress in public schools.
Hicks, 1942 [35] *	United States of Amerika	8 (8)	Boys 4 Girls 4	5 y 11 m–8 y 9 m	deaf
Johnson, 1975 [36]	United States of Amerika	182	Males 90 Females 92	10–19 y	Profoundly deaf; not benefit from hearing aids.
Johnson, 1977 [37]	United States of Amerika	133 (133)	Males 68 Females 65	11–19 y	deaf—not benefiting from hearing aids
Johnson and Khatena, 1975 [36]	United States of Amerika	417 (181)	Males 89 Females 92	10–19 y	profoundly deaf, i.e., they did not benefit from hearing aids
Kaltsounis, 1969 [38]	United States of Amerika	35	NS	second grade	deaf
Kaltsounis, 1970 [39]	United States of Amerika	777 (172)	NS	1st–6th grades	Deaf–hearing deprivation was substantial enough
Kaltsounis, 1970 [40]	United States of Amerika	418 (67)	NS	4th–6th grades	hearing deprivation was substantial enough to prevent them from making satisfactory progress in public schools
Kaltsounis, 1971 [41]	United States of Amerika	233	Boys 114 Girls 119	1st–4th grades	deaf
Laughton, 1988 [42]	United States of Amerika	28	Male 14 Female 14	8–10 y	85 dB loss or greater
Lubin and Sherrill, 1980 [43]	United States of Amerika	24 (12)	Boys 7 Girls 5	3–5 y	hearing loss (moderate, severe, profound)
Marschark and Clark, 1987 [44]	United States of Amerika	NS	NS	NS	NS
Marschark et al., 1987 [45]	United States of Amerika	NS	NS	NS	deaf
Marschark and West, 1985 [46]	United States of Amerika	8 (4)	Boys 3 Girl 1	12.10–15.0 y	≥80 dB
Marschark et al., 1986 [47]	United States of Amerika	40 (20)	Male 11 Female 9	8.1–14.8 y	deaf
Minarsih and Wahab, 2019 [48] *	Indonesia	1	NS	high school	deaf
Moorjhani et al., 1998 [49]	Rajasthan	80 (NS)	NS	6–11 y	from 55 dB to 89 dB
Pang and Horrocks, 1968 [50]	United States of Amerika	NS (11)	Boys 6 Girls 5	11–12 y	deaf
Paszkowska-Rogacz, 1992 [51]	Poland	44 (22)	Boys 10 Girls 12	13–15 y	deaf
Reber and Sherrill, 1981 [52]	United States of Amerika	10	Male 8 Female 2	9–14 y	71–90 dB 4 90 dB + 6
Silver, 1977 [53]	United States of Amerika	44 (22)	NS	NS	deaf
Stanzione et al., 2013 [22]	United States of Amerika	52 (17)	Male 10 Female 7	14–18 y	CI 3 hearing aids 14
Szobiová and Zborteková, 2006 [54]	Slovakia	69 (45)	Male 18 Female 27	18–88 y	NS
Yu et al., 2009 [55] *	People’s Republic of China	144 (122)	Male 62 Female 60	8–16	deaf

NS—non specified; y—years; m—months; dB—decibels; CI—Cochlear Implant. * Source Google Scholar

**Table 2 children-10-01383-t002:** Quantitative research—Study design, statistical methods, Aim and Tools.

	Study Design	Statistical Methods Used	Aim	Tools
Daramola et al., 2019 [30]	Case series	Split half, Cronbach’s Alpha, Spearman-Brown correlation	Difference between DHH and TH	a tool created by the authors Questionnaire on the Creativity Level of Students with Hearing- impaired and Hearing
Ebrahim, 2006 [32]	Case series	Multivariate analysis of variance (MANOVA)	Difference between DHH and TH	Torrance Tests of Creative Thinking-Figural, Form A (A1-3)
Ebrahim, 2006 [33]	Case series	Multivariate analysis of variance (MANOVA)	Difference between DHH and TH	Torrance Tests of Creative Thinking-Figural, Form A (A1-3)
Gallagher, 1968 [34]	Quasi-experimental Research Design	Split half and Kuder-Richardson coefficient of reliability	The relationship between Creativity thinking and IQ	The Abbreviated Form VII, Minnesota Tests of Creative Thinking. Stanford-Binet Intelligence Scales
Johnson, 1975 [36]	Quasi-experimental Research Design	Factor analysis	The relationship between creativity and onomatopoeic words	Onomatopoeia and Images, Form lB
Johnson, 1977 [37]	Quasi-experimental Research Design	Analysis of variance (ANOVA)	Difference between DHH and TH + intellectual function	Torrance Tests of Creative Thinking—Figural Form B;
Johnson and Khatena, 1975 [36]	Quasi-experimental Research Design	F-test, correlation	Difference between DHH and TH	Onomatopoeia and Images, Form 1B
Kaltsounis, 1970 [39]	Quasi-experimental Research Design	Analysis of variance (ANOVA)	Difference between DHH and TH	Torrance Tests of Creative Thinking-Figural, Form A
Kaltsounis, 1970 [40]	Quasi-experimental Research Design	Three two-way factorial analyses	Difference between DHH and TH	Torrance Test of Thinking Creatively With Words, Form A
Kaltsounis, 1971 [41]	Quasi-experimental Research Design	Two-way factor analysis	Level of creativity of DHH	Torrance Test of Thinking Creatively With Pictures, Form A (1966)
Laughton, 1988 [42]	Before and after studies	Multivariate analysis of variance (MANOVA)	Influence of creativity + two curricular designs	Torrance Tests of Creative Thinking
Lubin and Sherrill, 1980 [43]	Quasi-experimental Research Design	Analysis of covariance (ANCOVA)	Difference between DHH and TH + motor creativity	Torrance Tests of Thinking Creatively in Action and Movement
Pang and Horrocks, 1968 [50]	Quasi-experimental Research Design	Analysis of variance (ANOVA)	Difference between DHH and TH + intellect	Raven’s Coloured Progressive Matrices, Wallach and Kogan Creativity Test;
Paszkowska-Rogacz, 1992 [51]	Quasi-experimental Research Design	Medium values, standard deviation	Difference between DHH and TH	Barron–Welsh Art Scale, Torrance’s Tests of Creative Thinking, Wechsler Intelligence Scale for Children
Reber and Sherrill, 1981 [52]	Quasi-experimental Research Design	Pearson correlation, Student’s *t*-test and frequency analysis	Difference between DHH and TH + classroom behavior	The Test for Creative Thinking–Drawing Production; Raven’s Progressive Matrices test; Pupil Behavior Inventory
Stanzione et al., 2013 [22]	Quasi-experimental Research Design	Test-retest; Correlation; Covariations; F-test	Influence of creativity + dance/movement skills	Torrance Tests of Creative Thinking, Figural form B; Dance/Movement Skills Assessment
Szobiová and Zborteková, 2006 [54]	Quasi-experimental Research Design	Multivariate Analysis of Covariance (MANCOVA)	Difference between DHH and TH	Torrance Tests of Creative Thinking
Yu et al., 2009 [55]	Quasi-experimental Research Design	Student’s *t*-test	Difference between DHH and TH	General Ability Tests (Smith and Whetton)
Daramola et al., 2019 [30]	Quasi-experimental Research Design, Control study	*t*-test	Difference between DHH and TH	New Creativity Test, Raven’s Test

**Table 3 children-10-01383-t003:** Mixed Research—Design, Methods, Aim, and Tools.

	Research Design	Methods Used	Aim	Tools
Marschark and West, 1985 [46]	Phenomenological Studies + Quasi-experimental Research Design	Coding + Descriptive analysis and correlation.	Relationships between language and cognition and creativity	Story Production
Marschark et al., 1986 [47]	Phenomenological studies and Quasi-experimental Research Design	Coding + Descriptive analysis and correlation.	Description of the development of linguistic and cognitive flexibility and its impact on creativity	Story Production
Silver, 1977 [53]	Observation + Quasi-experimental Research Design	Thematic analysis + Descriptive analysis and correlation.	Cognitive skills and creativity skills	Torrance Tests of Creative Thinking

**Table 4 children-10-01383-t004:** Description of creativity of DHH individuals.

	Outputs	Verifiability of Research
Gallagher, 1968 [34]	There were indications that regarding means in numerical achievement tests, there were interactions between age, sex, and creativity.	Verified test + Cronbach’s alpha reliability
Kaltsounis, 1971 [41]	Mean fluency, flexibility, originality, and elaboration scores by grade level within racial groups showed a noticeable tendency for scores of white deaf children to exceed those for black children in divergent thinking only in the earlier grades, while the scores of the blacks tended to exceed those of the whites at the fourth-grade level.	Verified test + Verifiability exists
Marschark and Clark, 1987 [44]	The use of inappropriate research tools leads to children with hearing impairments being underestimated.	No verifiability
Marschark et al., 1987 [45]	Deaf children have a much greater tendency towards language flexibility than is normally expected.	No verifiability

**Table 5 children-10-01383-t005:** Possibilities for the development of creativity of DHH individuals.

	Outputs	Verifiability of Research
Arnidha and Hidayatulloh, 2019 [29]	From the number of five respondents, it turned out that one of the respondents was more creative, one was not creative, and the remaining three showed average creativity.	No verifiability
Davies, 1984 [31]	Exercises in creative dramatics promote flexibility of function and attitude in hearing-impaired students.	No verifiability
Johnson, 1975 [36]	The results show that the respondents who had caught the onomatopoeic words had a significantly higher mean score than the respondents who had not caught the words.	Verified tests sample, method + Verifiability exists
Kaltsounis, 1969 [38]	The results obtained indicate that the quality and quantity of the creative originality scores of these two groups of deaf children are not a sufficient basis for concluding that either method of instruction is superior.	No verifiability
Laughton, 1988 [42]	A positive influence on the development of creative thinking was shown in all parameters in the application of the Creativity Curriculum in 12 activities.	Verified tests, sample, methods + Verifiability exists
Minarsih and Wahab, 2019 [48]	Creativity was described as improved in the target group.	No verifiability
Reber and Sherrill, 1981 [52]	Dance training led to significant improvements in originality, elaboration, total creative thinking score, and dance/movement skills. The rationale for including the creative arts, particularly dance, in curricula for hearing-impaired students was confirmed as being effective.	Verifiability exists

**Table 6 children-10-01383-t006:** Differences in creativity between individuals with DHH and TH.

	Outputs	Verifiability of Research
Daramola et al., 2019 [30]	The creativity level of hearing-impaired students is essentially higher when contrasted with their hearing peers. The creativity level of females with hearing impairment is fundamentally higher than that of males. The finding further revealed that post-lingual hearing-impaired students have significantly higher creativity levels than their pre-lingual peers.	Reliability verified by split-half, Cronbach’s alpha
Ebrahim, 2006 [32]	The hearing children scored significantly (*p* < 0.05) higher than the deaf children in fluency, originality, and abstractness of titles. However, there were no significant differences between the deaf and hearing children in elaboration, resistance to premature closure, and creative strength (*p* < 0.05).	Clearly described validity and reliability
Ebrahim, 2006 [33]	The hearing children scored significantly (*p* < 0.05) higher than the deaf children in fluency, originality, and abstractness of titles. However, there were no significant differences between the deaf and hearing children in elaboration, resistance to premature closure, and creative strength (*p* < 0.05).	Verifiability exists
Halpin and Torrance, 1973 [28]	The deaf children studied by Kaltsounis (1970) scored markedly higher than the hearing children on verbal fluency and verbal originality. Blindness and deafness did not differentially affect the scores on the Torrance test for children in this study.	No verifiability
Hicks, 1942 [35]	Children with hearing impairments create different images (objects, nature), and intact children rather than animals, toys, or mixed objects. A problem was found especially with the description and creation of the story in children with hearing impairments. The children with hearing impairments worked with greater passion, were not critical of creative expression, and did not seek help and support. The hearing girls were significantly better than the deaf girls.	No verifiability
Johnson, 1975 [36]	The deaf children scored significantly higher than the hearing children on the Fluency, Flexibility, and Elaboration subtests of TTCT. The deaf children scored higher as their ages increased.	Verifiability exists
Johnson, 1977 [37]	The hearing respondents scored much higher than the deaf respondents.	Verifiability exists
Kaltsounis, 1970 [39]	The deaf children had better results than the hearing children in creative thinking abilities, except in grades 2 and 3 for fluency and grade 2 for originality and elaboration. The deaf and hearing males performed more poorly than the deaf and hearing females in figural fluency, flexibility, and elaboration. The opposite was the case in favor of the males for figural originality.	Verifiability exists
Kaltsounis, 1970 [40]	The older deaf children scored markedly higher than their hearing counterparts on fluency and originality but not on flexibility. In the case of verbal flexibility, the performance of the deaf children was equal to that of the hearing ones.	Verifiability exists
Lubin and Sherrill, 1980 [43]	The preschool deaf children were significantly inferior to the hearing children in motor creativity, as measured by the Torrance Test of Thinking Creatively in Action and Movement.	Verifiability exists
Marschark and West, 1985 [46]	The deaf children who used sign language used the same amount of figurative language as their hearing peers in English.	Verifiability exists
Marschark et al., 1986 [47]	The subjects here showed considerable use of creative language devices when evaluated in sign rather than vocal language. The present study, therefore, indicates that evaluating the linguistic and cognitive skills of deaf school children in terms of English underestimates their abstractive linguistic and cognitive abilities.	Reliability is described
Moorjhani et al., 1998 [49]	Comparison of intellect and creativity in both groups of respondents: in the RCPM test, the hearing children scored significantly higher, and creativity was not found to be a significant difference between the two groups in verbal items. The children with hearing impairments responded more to visual stimuli.	Verifiability exists
Pang and Horrocks, 1968 [50]	The deaf children scored lower than the hearing children on the Barron–Welsh Art Scale and Torrance Figural Tests of Creative Thinking. They were not interested in abstract figures but were more oriented toward concrete figures. In the Torrance dimensions, they scored about the same as the group of hearing children but were higher in processing.	Verifiability exists
Paszkowska-Rogacz, 1992 [51]	The level of creativity and other monitored items found in the tests was always lower in the deaf children than in the control sample of hearing children.	Verifiability exists
Silver, 1977 [53]	The deaf participants’ average scores (when compared to the hearing norms) were in the 88–99th percentile (i.e., originality, 99%; fluency, 97%; flexibility, 88%; and elaboration, 99%). The deaf participants’ pictures were judged to be more creative than those of the hearing participants.	Verifiability exists
Stanzione et al., 2013 [22]	The deaf students’ performance was equal to, or more creative than, that of the hearing students on the figural assessment of divergent thinking, but less creative on the verbal assessment.	Reliability is described
Szobiová and Zborteková, 2006 [54]	Differences were found only in some indicators of creative thinking: the adolescents with hearing impairment did not assign generalizing names to the drawings, they did not draw abstract themes and did not use symbols and signs in their drawings.	Verifiability exists
Yu et al., 2009 [55]	The deaf children were lower in verbal fluency, flexibility, originality, and figural flexibility. No difference was seen in figural fluency and originality.	Verifiability exists

## Data Availability

No new data were created or analyzed in this study. Data sharing does not apply to this article.

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
