# Peer review of "The Difference in the Creativity of People Who Are Deaf or Hard of Hearing and Those with Typical Hearing: A Scoping Review"

_children, 2023, doi:10.3390/children10081383_

Round 1
Reviewer 1 Report
The scoping review aimed to gather information on all available research in the field of the creativity of individuals with DHH. Overall, the design is comprehensive, and the included research evidence is of high quality. It systematically elaborates on various dimensions of creativity in individuals with DHH, providing a reference for future studies. However, there are some points that need further improvement.
1. Line 20. The first occurrence of "JBI methodology" in the abstract should be written in its full form, which is "Joanna Briggs Institute (JBI) methodology."
2. In section 1.1 of the introduction, it is not recommended to present it separately for introduction; instead, it should be integrated into the introduction section, and some of the introduction should be placed in the methods section.
3. The image quality of Chart 1 is not clear.
4. Line 197. “The extracted articles were written between 1942 and 2019”.--- The articles extracted and analyzed in this review were written between 1942 and 2019. It is recommended to conduct the extraction and analysis up to the most recent time available.
5. The discussion section only elaborates on the different dimensions of creativity in individuals with hearing impairments (DHH). It is recommended to delve into the possible underlying mechanisms and reasons behind these findings.
Author Response
Thank you for your helpful comments. Please see the attachment.

Reviewer 2 Report
This is a scoping review paper aimed with assessing current knowledge on differences in creativity depending on hearing status.
The title and abstract are fine. The introduction defined the aims of the study and the relevant background information needed to proceed with a scoping review. The main issue of the introduction is that it reads as a social sciences paper, and less as a methodologically driven medical field paper. There is nothing wrong with that, as it is well wrtitten and structured, but I would suggest avoiding subsections in the introduction and forming the last paragraph as an explicit aims/hypothesis section, while metodhodology (lines 60-66) belongs in the MM section, with PRISMA guidelines that need to be added to the Abstract. Deafness and hearing impairment need to be exactly defined using AAOHNS/ASHA-NIOSH criteria.
Lines 103-108: possibly consider prelingual deafness as a group. Its's unclear why people without pedagogical intervention were excluded. Creativity should not be dependent on outside intervention. Combining deaf and hard-of-hearing people is methodologically flawed, since people who are not deaf may hear reasonably well with hearing aids. Please commment.
Sources of unpublished studies/gray literature, including Clinical trials, Current controlled trials, CENTRAL, and Google Scholar, were also searched, which is problematic in terms of reproducible and rigorous results. I suggest labeling data stemming from Google.
Studies cited are quite old (over 40 years on average), so limitations regarding current concepts interpreted through outdated notions of hearing and cognitive processing should be underlined. Also, a wide range of studies did not specifically focus on children.
Since the studies are heterogeneous in methodology, focus and results, a clear and poignant results section is not possible. The results are further complicated by a threefold question line, where only the third applies to hearing issues. The tables are adequate and add to text flow and argument organization and presentation.
The discussion is well executed, and gives a succint overview of areas of creativity and DHH. Since the majority of studies have conflicting results, I would greatly value the information on what proportion of children included had severe or intermediate hearing loss, whether they were augmented by hearing aids and some modern data on hearing development and neural plasticity that might shed some light on creativity differences.
At present, the paper is speculative and does not offer substantial new data to warrant publication. However, if the authors make an additional effort and connect present knowledge on brain and hearing patophysiology with creativity development, I would support reconsidering the manuscript for publication.
Author Response

(The authors gave the same response as above.)

Reviewer 3 Report
Dear author,
The manuscript follows a scoping review methodology according to JBI.
The overall structure of the manuscript, and specifically Materials and Methods, should be organized in a logical sequence following the PRISMA-ScR results reporting guidelines. It is recommended to follow the PRISMA-Scr guidelines (this should be cited in the methods section (not only refers to flow diagram)).
In the abstract it is recommended not to use acronyms. The information on methodology should be expanded in the following order (including at least: design, databases, MesH terms, inclusion and exclusion criteria). It is more appropriate to present the methodology first and then the results.
Keywords: For indexing the article correctly (if the manuscript is accepted) MeSH terms must be used. The following keywords not adjust to MeSH term: Deaf, "Hard of hearing", "Typical hearing", "Fluency", "Flexibity", "Scoping review".
"Originality" is an Entry term for the MeSH "Creativity".
"Hearing Loss" (definition: A general term for the complete or partial loss of the ability to hear from one or both ears) is an example for a possible term to be used.
Introduction: It is well written, from general to specific aspects. On page 2, lines 58-59, they should describe the acronym "IQ" in its first use.
The last paragraph of the introduction (before point 1.1), lines 60-67, corresponds to methods section.
Aims: The aim (and research question) should be described at the end of the introduction.
Materials and Methods: Should start with study design.
Was any review protocol previously published on any platform such as OSF or similar? Reference number 25 is not well referenced (accepted for publishing).
In the participants section, the inclusion and exclusion criteria are not clearly defined in relation to the population. Although the criteria are suggested, they should be expressed more clearly. It is not clear that only children have been included as inclusion criteria (in Table 1 only one study includes studies over 18 years of age, most of them are in children and some include adolescents). This aspect is important to clarify.
Point 2.2 Methods does not apply (it is already included in the Materials and methods section). See previous comments on the abstract.
The complexity of the scoping review performed, together with the fact that the search terms have not been clarified, requires that the search strategies be reported. In the search strategy section it is recommended to identify the strategies used in each database, search dates, Boolean operators, MeSH terms and free terms if they were used. The following sentence is too unspecific to define this aspect: "The words contained in the titles and abstracts of relevant articles and the index terms used to describe the articles were used to develop a full search strategy. The search strategy, including all identified keywords and index terms, was adapted for each database and/or information source that is included". The search dates must be specified. In this section it is advisable to report a table or supplementary material with this information.
Flow chart is the first result of the review, it should be moved to the results section. The quality of the flow chart resolution should be improved.
In the data extraction section, the review question should not be repeated. it is sufficient to identify the research outcomes.
Results: Review the format of the tables according to the Children's Template.
Check in all tables the first column: Column headings and correctly identify year of publication.
In Table 1, NS, PRC, y, dB, CI acronym? (to clarify all acronyms at the foot of the table). Authors: Stanzione et al [23], how many male and female?
Table 2, 3, and Table 4, 5: It is possible to unify the tables in one table for 2, 3 and 4, 5?
In section 3.2.3 Outputs, the criterion followed is not clear. It is understood that during the screening process to select the studies to be included in full text, a critical appraisal process has been carried out, using the JBI tools specific to each design, to include studies that meet the quality criteria established by the authors. Verifiability of research in the following tables is not understood, since all the studies should have passed this criterion and the corresponding specific data should be extracted from these tables.
Author Response

(The authors gave the same response as above.)

Reviewer 4 Report
The following modifications are required in the article:
• Specify title in correspondence with the proposed objectives.
• Improve the summary including the main objective, method used, most relevant results and discussion.
• In the introduction section, it could be improved in terms of theoretical depth:
o Exposing some results obtained in recent research on creativity in hearing and deaf subjects.
o Specifying in the justification about the importance and necessity of carrying out the research that is presented.
• It is necessary to include the objectives of the investigation, as well as to explain and justify the adopted investigation method, as well as to deepen in a concrete way the procedure that has been developed.
• In the results section, it is necessary to interpret in detail the data obtained in relation to the objectives to be set. In addition, it is necessary to interpret the most relevant data in the tables.
• In the discussion section, it would be advisable to compare the most relevant results in this research with those obtained in other previous studies with similar themes.
• In the conclusion section, it is convenient to present the most relevant conclusions in response to the proposed objectives, as well as include the implications of the study carried out for educational practice.
Author Response

(The authors gave the same response as above.)

Round 2
Reviewer 1 Report
Accept in present form
Author Response
Thank you very much for your time and work for us.
Reviewer 2 Report
The authors have responded to the main issues raised in the manuscript well. I would recommend publication in light of the revisions made.
Minor spell-checking required.
Author Response
Dear reviewer, thank you very much for your time and comments. The spelling is now corrected.
Reviewer 3 Report
Dear Authors,
thank you for the replies and explanations provided.
In the tables you have included the year together with the author in the first column; however, you have deleted the citation in this new version, which should also be included (eg: Arnidha & Hidayatulloh; 2019 [29]).
Previous report comment: Stanzione et al [23], how many male and female? your letter response was Male: 10; Female: 7; but this date is not implemented in the table. Thus, Kaltzounis 1969, 1970a, 1970b and Moorjhani et al 1998 non specified number of male/female. If no date possible explain you should describe NS.
Regarding the location of the flowchart the authors can consult the sentence: "The results section of a scoping review could be considered to contain two broad sections, the first of which describes the results of the search strategy and the selection process, including a PRISMA flowchart" in the following JBI reference: https://journals.lww.com/jbisrir/Fulltext/2020/10000/Updated_methodological_guidance_for_the_conduct_of.4.aspx?context=FeaturedArticles&collectionId=5
The JBI template for a scoping review is also available at the following link: https://jbi.global/scoping-review-network/resources
Author Response
Dear reviewer, thank you for the additional recommendation and review (spelling corrected - thanks). Always such recommendations are valuable for our further development (though in the beginning, when we get reviews ...).
